# Clinical Study of the Relationship between Sjögren Syndrome and T-Cell Large Granular Lymphocytic Leukemia: Single-Center Experience

**DOI:** 10.3390/ijms232113345

**Published:** 2022-11-01

**Authors:** Vadim Gorodetskiy, Vladimir Vasilyev, Yulia Sidorova, Bella Biderman, Natalia Kupryshina, Murad Vagida, Natalya Ryzhikova, Andrey Sudarikov

**Affiliations:** 1Department of Intensive Methods of Therapy, V.A. Nasonova Research Institute of Rheumatology, 115522 Moscow, Russia; 2Joint and Heart Treatment Center, 107140 Moscow, Russia; 3Laboratory of Molecular Hematology, National Medical Research Center for Hematology, 125167 Moscow, Russia; 4Hematopoiesis Immunology Laboratory, Russian Cancer Research Center N.N. Blokhin, 115478 Moscow, Russia; 5Laboratory of Transplantation Immunology, National Medical Research Center for Hematology, 125167 Moscow, Russia

**Keywords:** Sjögren syndrome, T-cell large granular lymphocytic leukemia, STAT3 mutation, next-generation sequencing, immunophenotyping

## Abstract

The relationship between Sjögren syndrome (SS) and T-cell large granular lymphocytic (T-LGL) leukemia remains unclear. In this paper, we report for the first time a large case series of 21 patients with primary and secondary SS associated with T-LGL leukemia. Our results suggest the importance of considering T-LGL leukemia in the diagnostic evaluation of SS patients, particularly when neutropenia occurs. We also postulate that elevated antinuclear antibody titers in patients with T-LGL leukemia indicate the need for the clinical assessment of SS. To assess whether SS affects the frequency of the signal transducer and activator of transcription 3 (*STAT3*) gene mutations in T-LGL leukemia, we examined *STAT3* mutations by next-generation sequencing in two cohorts of patients: with SS-associated T-LGL leukemia and T-LGL leukemia in the setting of rheumatic diseases but without SS. While our results suggest that SS, per se, is not associated with an increased frequency of *STAT3* mutations in T-LGL leukemia, further studies are needed to better assess the role of the STAT pathway in the development of concomitant SS and T-LGL leukemia.

## 1. Introduction

Sjögren syndrome (SS) is a multisystem autoimmune rheumatic disease (RD) that can occur in isolation (defined as primary SS (pSS)) or as a complication of or an overlap with other rheumatic conditions (defined as secondary SS (sSS)) [1,2,3]. Of the various autoimmune RDs, patients with SS have the highest risk of developing lymphoma [4]. The majority of lymphomas in the setting of SS are of B-cell origin [5], whereas T-cell lymphomas are rare [6,7].

T-cell large granular lymphocytic (T-LGL) leukemia is a rare chronic lymphoproliferative disease characterized by the clonal expansion of T cells that are larger in size than most circulating lymphocytes and contain azurophilic granules in the cytoplasm [8]. Typical manifestations of T-LGL leukemia include LGL lymphocytosis, neutropenia, and splenomegaly [9]. The neoplastic cells typically express CD3, CD8, and CD57 with or without CD16; contain T-cell-restricted intracellular antigen 1 (TIA-1) and granzyme B; and have diminished or absent expression of the pan-T-cell markers CD5 and/or CD7 [10,11].

A peculiar feature of T-LGL leukemia is its association with autoimmune disorders, most commonly with rheumatoid arthritis (RA) [8,9]. In contrast, few cases of SS in the setting of T-LGL leukemia have been reported in the literature, and the pathogenic relationship between SS and T-LGL leukemia is unclear. Ramos et al. demonstrated the involvement of abnormal signal transducer and activator of transcription 3 (STAT3) signaling, apparently as a result of impaired inactivation of phosphorylated STAT3 (pSTAT3) protein in the T lymphocytes of patients with pSS [12]. Activation of the STAT3 pathway leads to decreased apoptosis and, thus, can result in the expansion of autoreactive T cells [12,13]. Perhaps the persistence of autoreactive T lymphocytes, probably in combination with other factors, leads to increased risk for the occurrence of activating point mutations of the *STAT3* gene, which is considered a hallmark of T-LGL leukemia [14].

The characteristics of SS-associated T-LGL leukemia have not yet been studied. We sought to address this by studying a large cohort of patients with SS-associated T-LGL leukemia, in which we examined the immunophenotypic and molecular characteristics of this pathology.

## 2. Results

Of the 21 patients with SS-associated T-LGL leukemia, pSS-associated T-LGL leukemia was diagnosed in 3 (14%) patients, and sSS-associated T-LGL leukemia was diagnosed in 18 (86%) patients. sSS was associated with RA in 16 of these 18 patients and with systemic lupus erythematosus (SLE) in the remaining 2 patients. The characteristics of the 21 patients with SS-associated T-LGL leukemia are shown in Table 1.

The median age of the patients at the time of T-LGL leukemia manifestation was 55 years (range, 36–72 years). The median age of patients at the time of SS manifestation could not be determined because the exact dates for the onset of SS clinical manifestations were lacking for the majority of patients.

Neutropenia < 1.5 × 10^9^/L (median 0.610 × 10^9^/L, range, 0.000–1.479 × 10^9^/L) was detected in all of the patients. LGL counts of ≥2 × 10^9^/L in the peripheral blood were found in 2 (12%) of 17 patients whose blood smears were available. Splenomegaly was observed in nine (43%) patients.

The immunophenotype of the tumor cells was examined in 20 patients: by flow cytometry of fresh peripheral blood mononuclear cells (PBMCs) in 19 patients and by immunohistochemistry in spleen in 1 patient.

CD8 expression on CD3+ T-lymphocytes was bright in 18 cases and diminished in 2 cases. In one patient with diminished membrane expression of CD8, these cells were also weakly positive for CD4 (Figure 1).

Cytotoxic (CD3+/CD8low or CD3+/CD8+) T-lymphocytes coexpressed CD57+ in 16 (80%) of 20 cases and CD16 in 6 (32%) of 19 cases. Diminished (or absent) membrane expression of CD5 on CD8low/+ lymphocytes was detected in 18 (90%) of 20 cases. Overall, phenotypic abnormalities characteristic for T-LGL leukemia were detected in 19 of the 20 studied patients.

Evaluation for T-cell receptor (*TCR*) genes rearrangement was investigated in all 21 patients: in peripheral blood in 20 patients, and in spleen tissue in 1 patient. Monoclonal rearrangement of *TCR* genes was detected in 20 patients (in 19 patients in peripheral blood and in 1 patient in spleen tissue), and polyclonal rearrangement of *TCR* genes was observed in 1 patient, in the peripheral blood.

In the cohort of T-LGL leukemia with SS, *STAT3* mutations were found in 12 (57%) patients: in 1 patient with pSS and in 11 patients with sSS. Mutational hot spots included D661Y in seven cases; N647I in three cases; and Y640F, S614R, D661V, K658R, N664T, and S614G in one case each (Figure 2). Three out of the twelve patients had more than one *STAT3* mutation.

To assess whether SS affects the frequency of *STAT* gene mutations in T-LGL leukemia, we examined *STAT3* mutations in a cohort of 45 patients with T-LGL leukemia and RD but without SS (Appendix A). The *STAT3* mutation frequency was not different in the cohorts of T-LGL leukemia with and without SS, totaling 57% (12 of 21 patients) in the cohort of patients with SS and 60% (27 of 45 patients) in the cohort with RD but without SS (*p* = 1).

In the case in which coexpression of CD4low and CD8low was found, *STAT3* mutations and T-cell clonality were evaluated on the sorted cell populations: CD4low and CD4high (control). *STAT3* mutations and T-cell clonality were detected only in the CD4low fraction (Figure 3 and Table 1).

Integration of the typical immunophenotype with T-cell clonality and clinical manifestations allowed us to diagnose T-LGL leukemia in 20 patients according to the criteria given by Matutes [15]. In one patient with a polyclonal rearrangement of the *TCR* gene in the blood, we diagnosed T-LGL leukemia based on typical clinical findings and the presence of the *STAT3* mutation highly specific to T-LGL leukemia [16].

A positive Schirmer test (in either eye of <5 mm/5 min) and/or abnormal Rose Bengal ocular surface staining were found in 13 of 21 patients. Radiological or ultrasound evidence of glandular parenchymal abnormalities characteristic of SS was observed in 17 of 19 examined patients (in 2 patients, these studies were not performed). Anti-Ro/SSA antibodies were detected in seven patients. A combination of antinuclear antibodies (ANA) ≥1:320 with positive rheumatoid factor (RF) was detected in 18 patients.

All 21 patients in our cohort met the two criteria for the diagnosis of SS described by Baer [17]: (1) having objective markers of dry eye and/or salivary gland parenchyma abnormalities characteristic of SS detected using imaging techniques, and (2) serologic evidence of SS such as anti-Ro/SSA antibodies (with or without anti-La/SSB antibodies) or, alternatively, a combination of an ANA ≥1:320 with a positive RF. Focal lymphocytic sialadenitis of the minor salivary gland was diagnosed in six of seven patients.

## 3. Discussion

Apparently, the first description of SS with LGL leukemia was presented by Goske et al. (1980), who reported on a 52-year-old woman with SS, absolute lymphocytosis, and neutropenia [18]. We found only 39 cases of T-LGL leukemia in SS patients in PubMed to date [7,19,20,21,22,23,24,25,26,27,28]. Of these, 31 cases of T-LGL leukemia were associated with pSS, 2 cases with sSS (one patient had RA and another had limited scleroderma), and in 6 cases, no information was provided.

The largest case series of SS-associated T-LGL leukemia was reported by Friedman et al. [24]. Of the 48 patients presenting with T-LGL leukemia, 13 (27%) were diagnosed with SS (12 patients had pSS, and 1 had sSS). In contrast, the French registry of 201 patients with T-LGL leukemia reported only 4 (2%) patients with SS [26]. This difference in the prevalence of SS in T-LGL leukemia could be explained by the fact that in the cohort reported by Friedman et al., the researchers interviewed patients for clinical signs of SS. Manifestations of SS are usually mild in comparison with RA symptoms, and SS may not be diagnosed unless specifically sought [29].

It is worth noting that according to the results of the various studies shown in Table 2, the frequency of ANA and RF in patients with T-LGL leukemia is significantly higher than the frequency of diagnosed RA.

The mismatch between the frequency of serological findings and the clinical diagnosis of RD is usually interpreted as a consequence of B-cell dysfunction in patients with T-LGL leukemia [34]. However, the combination of a positive ANA test with a positive RF test is a serological sign of SS [17]. As noted by Friedman et al., most patients in their T-LGL leukemia cohort did not report sicca symptoms unless specifically asked [24]. Considering the above, it is presumed that some patients diagnosed with T-LGL leukemia also had undiagnosed SS.

In our cohort of 85 patients with RD-associated T-LGL leukemia, as well as in the study by Friedman et al., patients were specifically interviewed to identify clinical symptoms of SS. If clinical or laboratory manifestations of SS were present, a comprehensive examination was performed. Our results on the incidence of SS in patients with T-LGL leukemia are consistent with the data obtained by Friedman et al.: 32% (21 of 66 patients) vs. 27% (13 of 48 patients) of patients, respectively.

It is hypothesized that an antigen-driven mechanism represents an initial step in LGLs expansion [35,36]. In cases of SS-associated T-LGL leukemia, it can be speculated that prolonged stimulation of an unknown antigen localized in the salivary and/or lacrimal glands, under certain conditions, can act as the triggering event for T-LGLs proliferation. Immunohistochemical analysis of salivary glands in the patients with pSS showed that CD4+ T cells were the most common cell type, but about 20% of the infiltrating cells were CD8+ T lymphocytes, which contribute to acinar damage [37,38]. In addition, Molad et al. observed the infiltration of minor salivary glands by cells with the T-LGL leukemia phenotype in a patient with pSS, neutropenia, and bone marrow involvement [22].

T-LGL leukemia is typically a disorder of CD8+ cytotoxic T cells, whereas the CD4+ (with or without CD8 coexpression) phenotype of T-LGL leukemia is uncommon. For example, out of 295 patients with T-LGL leukemia, only 5 patients (1.7%) had the CD4+ phenotype [33]. Similarly, in a study by Zhu et al., only 1 (0.9%) of 108 patients with T-LGL leukemia had the CD4+/CD8− phenotype [32]. Furthermore, CD4+ T-LGL leukemia, unlike CD8+ T-LGL leukemia, is not associated with autoimmune events. With the exception of two patients reported by Covach et al. [39], none of the 58 patients in all the studies for which clinical information was provided had RD [33,40,41,42]. Surprisingly, Friedman et al., in a cohort of 13 patients with SS-associated T-LGL leukemia, observed clonal dominant expansion of CD4+ T-lymphocytes in 4 (31%) patients [24]. Moreover, out of five patients with pSS-associated T-LGL leukemia, four patients had the typical CD8+ phenotype [25,27,28], and in one patient T-LGLs coexpressed CD4 and CD8 [7].

In our cohort, CD8+ phenotype of T-LGL leukemia was dominant, but in one pSS patient, coexpression of CD4low and CD8low was observed in leukemic cells.

Given that clinical manifestations of SS are sometimes mild, it can be presumed that pSS is not diagnosed in some patients with CD4+ T-LGL leukemia. However, all 34 patients with CD4+ T-LGL leukemia reported in Lima et al. [40] and 5 of the 8 examined patients reported in Olteanu et al. [41] were negative for RF and ANA, making the possibility of undiagnosed SS unlikely. Further data are required to verify whether the CD4+ phenotype of T-LGL leukemia is associated with pSS.

Constitutive activation of the STAT3 pathway is a hallmark of T-LGL leukemia pathobiology [13,43], and *STAT3* mutations were detected in 27–72% of patients with T-LGL leukemia [13,44,45,46]. To the best of our knowledge, *STAT3* mutation studies have only been performed in three patients with SS-associated T-LGL leukemia [27]. In all of these cases, no *STAT3* mutations were detected. However, the oligoclonal expansion of LGLs was detected in all of these patients.

Several studies have shown that patients with *STAT3* mutated status are more likely to have rheumatoid arthritis [13,44], but the association between other autoimmune diseases (including SS) and the frequency of *STAT3* mutations in patients with T-LGL leukemia has not been studied.

In our study, the frequency of RA was similar in the cohort of T-LGL leukemia with SS and in the cohort of T-LGL leukemia with RD but without SS: 76% (16 of 21 patients) and 84% (38 of 45 patients), respectively (*p* = 0.6404). This similarity counterbalanced the effect of RA on the results of the frequency of *STAT3* mutations in both cohorts. Although our results suggest that SS, per se, is not associated with an increased frequency of activating point mutations in the *STAT3*, we think that a study of *STAT3* mutations in a cohort of pSS patients without sign of T-LGL leukemia would clarify this issue.

## 4. Patients and Methods

A total of 124 patients over 18 years of age were admitted to the V.A. Nasonova Research Institute of Rheumatology from January 2008 to May 2022 with various RDs and suspected LGL leukemia. LGL leukemia was suspected if the patient had unexplained neutropenia <1.5 × 10^9^/L or LGL lymphocytosis. The diagnosis of LGL leukemia was confirmed in 87 patients: 2 patients had NK-cell LGL leukemia, and 85 patients had T-cell LGL leukemia. All 85 patients with RD-associated T-LGL leukemia were interviewed by a rheumatologist with the aim of detecting the clinical manifestations of SS. For 19 of these 85 patients, examination data for the diagnosis of SS were not available. SS was absent in 45 (68%) of the remaining 66 patients and confirmed in 21 (32%) patients. The data from these 21 patients with SS-associated T-LGL leukemia are presented in this paper.

### 4.1. Evaluation of T-Cell Clonality

The peripheral blood of 20 patients was collected in tubes with ethylenediaminetetraacetic acid (EDTA). Deoxyribonucleic acid (DNA) was isolated essentially as described by Miller et al. [47]. In one case (patient #13), a peripheral blood sample was unavailable and DNA was extracted from formalin-fixed paraffin-embedded (FFPE) tissue of the spleen as described earlier [48]. Rearranged *TCR-γ* (Vγ–Jγ) and *TCR-β* (Vβ–Jβ, Dβ–Jβ) were PCR-amplified using multiplex primer sets essentially as described by van Dongen et al. [49]. Amplified fragments were detected using an ABI PRISM 3130 Genetic Analyzer (Applied Biosystems, Foster City, CA, USA). The data were analyzed using GeneMapper software version 4.0 (Applied Biosystems).

### 4.2. Evaluation of STAT3 Mutations

In the cohort of 21 patients with T-LGL leukemia and SS, *STAT3* mutations were examined using genomic DNA extracted from specimens of the peripheral blood (20 patients) and spleen (patient #13).

In the cohort of 45 patients with T-LGL leukemia and RD but without SS, *STAT3* mutations were examined using DNA extracted from specimens of the peripheral blood (43 patients), bone marrow (11 patients), and spleen (5 patients) (Appendix A).

Mutations in exons 19–21 of the *STAT3* gene were identified by NGS. Appropriate DNA regions were amplified using primers for exons 19–20 (product length 502 bp) and exon 21 (product length 522 bp), as previously described [13].

Amplified DNA fragments were converted into sequencing libraries using Nextera XT DNA Library Prep and a Nextera XT Index Kit v2 (Illumina, San Diego, CA, USA) according to the manufacturer’s instructions. Nucleotide sequences were analyzed on a MiSeq sequencer (Illumina) using a MiSeq Reagents Kit v2 for 300 cycles (Illumina). Raw data filtering, trimming of accessory sequences, alignment to the reference, and variant calling were performed using Trimmomatic [50], BWA [51], SAMtools [52], and VarDict [53] software. Usually, 2000–5000 reads were obtained for each target, and the cutoff for reporting variants was set to 0.5%. Discovered variants were annotated with the ANNOVAR [54] utility using the snp138 [55], refGene [56], ClinVar [57], and COSMIC [58] open databases.

### 4.3. Flow Cytometric Immunophenotypic Analysis

Fresh PBMCs were isolated using density gradient centrifugation on Ficoll-Paque (Pharmacia, Uppsala, Sweden) from 19 EDTA peripheral blood specimens. In two cases (patients #13 and #16), blood samples were not available for flow cytometric immunophenotypic analysis. Flow cytometric immunophenotyping was performed on a BD FACSCanto™ II system (Becton Dickinson, San Jose, CA, USA) using FCS Express version 3 software (De Novo Software, Los Angeles, CA, USA). Approximately 100,000 events were analyzed in each sample in the initial forward scatter (FSC) and side scatter (SSC) dot plot. The debris and doublets were excluded based on the FSC/SSC profiles. Furthermore, lymphocytes were gated using CD45 and side scatter. The following monoclonal antibodies from BD Biosciences (San Jose, CA, USA) were used: CD3, CD4, CD5, CD7, CD8, CD16, CD19, CD45, and CD57.

### 4.4. Fluorescence-Activated Cell Sorter

In one case (patient #20), cells were sorted on a FACS Aria III cell sorter (BD Biosciences) after staining with fluorescence-labeled monoclonal antibodies recognizing CD4 and CD45. The sorting procedure was performed using a 70 μm nozzle. Cells were washed once after sorting and resuspended in phosphate-buffered saline for downstream analysis.

### 4.5. Immunohistochemistry Analysis

Immunohistochemistry was performed on an FFPE spleen sample from patient #13. The following antibodies were used at dilutions suggested by the manufacturers: CD3 (polyclonal, Dako, Carpinteria, CA, USA); CD4 (clone 4B12, Dako); CD8 (clone C8/144B, Dako); CD16 (clone 2H7, Novocastra Laboratories, Newcastle upon Tyne, UK); CD20 (clone L26, Dako); and TIA-1 (clone 2G9, Immunotech, France). Immunostaining was performed using an Autostainer Link 48 (Dako, Denmark) according to the manufacturer’s instructions. All immunostained samples were counter-stained with hematoxylin.

### 4.6. Evaluation of Patients for SS

Patients were tested to diagnose dry eye (Schirmer test and Rose Bengal staining) and salivary gland parenchymal anomalies (parotid gland contrast sialography or salivary gland ultrasonography). Histologic examination of the labial salivary gland was performed in 7 patients.

Immunologic tests for the evaluation of SS were performed in all 21 patients. They included detection of precipitating antibodies against the extractable nuclear antigens Ro/SS-A and La/SS-B by ELISA, the determination of ANA using an indirect immunofluorescence assay on Hep-2 cells, and the characterization of RF by nephelometry.

### 4.7. Statistical Analysis

Descriptive statistics are presented as numbers and percentages for categorical data and as medians and ranges for continuous data. A two-sample test for equality of proportions with continuity correction was used for the statistical analysis. *p* < 0.05 was considered to indicate statistically significant differences.

## 5. Conclusions

Our results provide evidence of the importance of considering T-LGL leukemia in the diagnostic assessment of patients with SS, particularly when neutropenia occurs. We also postulate that in patients with T-LGL leukemia and increased ANA titers, subclinical manifestation of SS must be excluded. While our results suggest that SS, per se, is not associated with an increased frequency of *STAT3* mutations in T-LGL leukemia, further studies are needed to more thoroughly assess the role of the STAT pathway in the development of concomitant SS and T-LGL leukemia.

## Figures and Tables

**Figure 1 ijms-23-13345-f001:**
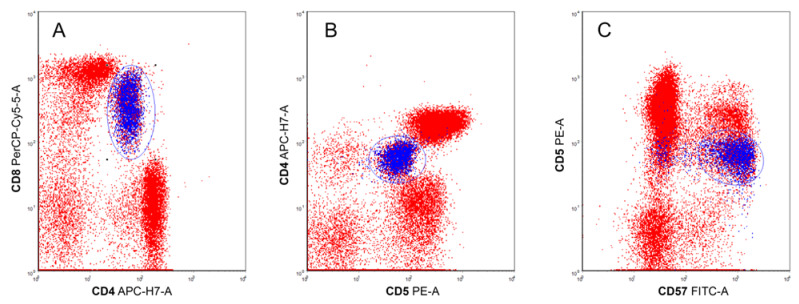
Case #20. Flow cytometry of T-cell large granular lymphocytic leukemia in peripheral blood shows (**A**) CD4low/CD8low (depicted in blue), (**B**) CD4low/CD5low (depicted in blue), and (**C**) CD5low/CD57+ (depicted in blue) expression.

**Figure 2 ijms-23-13345-f002:**
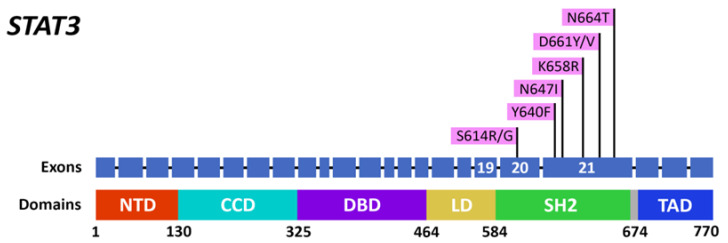
Distribution of *STAT3* mutations in the SH2 domain detected within patients with T-LGL leukemia and Sjögren syndrome.

**Figure 3 ijms-23-13345-f003:**
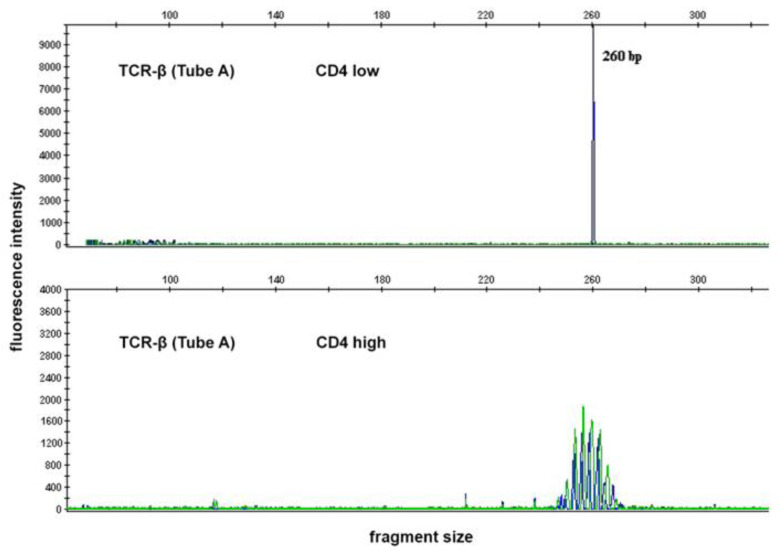
Case #20. Fragment analysis of T-cell receptor beta (*TCR-β*) gene PCR products from the CD4low fraction shows a clonal rearrangement pattern; in contrast, the CD4high fraction shows a polyclonal rearrangement pattern.

**Table 1 ijms-23-13345-t001:** Characteristics of 21 patients with SS-associated T-LGL leukemia.

Patient no./Sex/Age (y) *	Diagnosis	Positive Schirmer Test and/or Abnormal Ocular Surface Staining	Structural Abnormalities of Salivary Glands Characteristic of SS Detected by Imaging	Focal Lymphocytic Sialadenitis of Minor Salivary Glands	Anti-Ro/Anti-La	RF/ANA	Neutrophil Count (×10^9^/L)	LGL Count (×10^9^/L)	Splenomegaly	Samples for Testing/T-Cell Clonality/*STAT3* Mutations	Immunophenotype
1./F/60	RA and sSS	+	+	ND	+/−	+/+	0.812	0.479	+	PB/+/N647I	CD3+, CD8+, CD57+, CD5low/−, CD16+
2./F/43	RA and sSS	+	−	+	+/+	+/+	1.160	ND	−	PB/+/−	CD3+, CD8+, CD57+, CD5low/−, CD16−
3./F/64	RA and sSS	−	+	ND	−/−	+/+	0.468	3.900	−	PB/+/D661Y	CD3+, CD8+, CD57+, CD5low/−, CD16−
4./F/36	RA and sSS	+	+	ND	−/−	+/+	0.126	1.029	+	PB/+/D661Y	CD3+, CD8+, CD57+, CD5low/−, CD16−
5./F/59	RA and sSS	−	+	ND	−/−	+/+	0.975	1.482	−	PB/+/ N647I; Y640F; S614G	CD3+, CD8+, CD57+, CD5low/−
6./F/47	RA and sSS	−	+	−	−/−	+/+	0.495	1.215	−	PB/+/S614R	CD3+, CD8+, CD57+, CD5low/−, CD16−
7./F/57	RA and sSS	−	+	ND	−/−	+/+	0.000	0.625	+	PB/+/D661V; D661Y	CD3+, CD8+, CD57+, CD5low/−, CD16−
8./F/62	RA and sSS	+	+	ND	−/−	+/+	0.440	1.380	+	PB/+/−	CD3+, CD8+, CD57+, CD5low/−, CD16+
9./F/48	RA and sSS	+	+	ND	−/−	+/+	0.369	2.747	+	PB/+/N647I	CD3+, CD8+, CD57+, CD5low/−, CD16+
10./F/61	RA and sSS	+	+	ND	−/−	+/+	0.196	0.028	+	PB/+/D661Y	CD3+, CD8+, CD57+, CD5low/−, CD16−
11./F/36	RA and sSS	−	+	ND	−/−	+/+	0.410	ND	+	PB/+/D661Y	CD3+, CD8+, CD57+, CD5low/−, CD16+
12./F/51	RA and sSS	+	ND	ND	−/−	+/+	0.610	0.095	+	PB/+/−	CD3+, CD8+, CD57+, CD5low/−, CD16+
13./F/63	RA and sSS	+	ND	ND	−/−	+/+	0.056	ND	+	Spleen/+/−	IHC: CD3+, CD8+, CD57−, CD5−, TIA-1+, CD16−
14./F/72	RA and sSS	−	+	+	−/−	+/+	0.630	1.661	−	PB/+/−	CD3+, CD8+, CD57+, CD5+, CD16−
15./F/55	RA and sSS	+	−	ND	−/−	+/+	0.840	0.720	−	PB/+/−	CD3+, CD8low, CD57+, CD5low/−, CD16+
16./F/46	RA and sSS	+	+	ND	−/−	+/+	1.326	ND	−	PB/−/D661Y	ND
17./F/56	SLE and sSS	−	+	+	+/−	−/+	1.085	0.413	−	PB/+/K658R	CD3+, CD8+, CD57−, CD5−, CD16−
18./F/50	SLE and sSS	+	+	+	+/−	−/+	1.272	0.456	−	PB/+/−	CD3+, CD8+, CD57+, CD5low/−, CD16−
19./F/62	pSS	+	+	ND	+/−	−/+	1.479	1.377	−	PB/+/−	CD3+, CD8+, CD57−, CD5low/−, CD16−
20./F/51	pSS	+	+	+	+/+	+/+	1.029	0.441	−	PB/+/N664T; D661Y	CD3+, CD4low, CD8low, CD5low, CD57+, CD16−
21./F/54	pSS	−	+	+	+/+	+/+	0.594	0.308	−	PB/+/−	CD3+, CD8+, CD57−, CD5+, CD16−

*, at the time of detection neutropenia, lymphocytosis, or splenomegaly; y, years; RF, rheumatoid factor; ANA, antinuclear antibodies; +, positive/present; −, negative/absent; ND, no data; LGLs, large granular lymphocytes; *STAT*, signal transducer and activator of transcription gene; PB, peripheral blood; IHC: immunohistochemistry.

**Table 2 ijms-23-13345-t002:** Frequency of antinuclear antibodies, rheumatoid factor, and rheumatoid arthritis reported in T-LGL leukemia studies.

	Loughran [9]	Dhodapkar et al. [20]	Semenzato et al. [30]	Neben et al. [31]	Bareau et al. [26]	Zhu et al. [32]	Dong et al. [33]
Antinuclear antibodies	38% (19/50)	44% (22/50)	38% (26/69)	48% (11/23)	48% (34/70)	45%	21.9% (25/114)
Rheumatoid factor	57% (34/60)	61% (24/39)	43% (35/82)	48% (10/21)	41% (30/72)	10%	38.7% (41/106)
Rheumatoid arthritis	28% (36/129)	26% (18/68)	Not available	20% (9/44)	17% (35/201)	3%	11.9% (38/319)

## Data Availability

All the data presented in this study are available in this manuscript.

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
