# Peer review of "Clinical Study of the Relationship between Sjögren Syndrome and T-Cell Large Granular Lymphocytic Leukemia: Single-Center Experience"

_ijms, 2022, doi:10.3390/ijms232113345_

Round 1

Reviewer 1 Report

The manuscript by V. Gorodetskiy et al. describes clinical, molecular and immunophenotypic characteristics of peripheral blood samples or spleen sections from a small cohort of patients with Sjogren’s syndrome and T-cell large granular lymphocytic leukemia. The majority of samples were CD3+, CD8+, CD57+, and CD5 low/-; a minority also expressed CD16. Monoclonal rearrangement of the TCR beta chain was detected in 20 out of 21 patients and one or more STAT3 mutations in 12 patients. The Investigators compare their findings to the limited existing published data on patients with SS and T-LGL.

Overall, the manuscript adds information on a disease for which little data are available in the literature.

Principal comments:

The title of the manuscript is very general and implies that it presents a review on a well-characterized relationship. It should be modified to reflect the fact that the study presents original data on a topic that requires further investigation.

Page 2, line 69: It would be interesting to know what type of rheumatic disease was diagnosed in the 45 T-LGL patients who did not have SS.

On page 9, first paragraph, does the p value refer to a statistical analysis of the difference in percentages of patients with pSS in the present study and the study by Friedman et al.? Such a statistical comparison is does not add useful information without knowledge of Friedman et al.’s study design and patient cohort characteristics.

The Discussion section describes STAT3 mutation and RA frequencies for an additional 45 T-LGL patients without SS. The manuscript would be strengthened by including data on these patients as reported in Table 1 for the 21 SS patients.

Minor comments:

The description of evaluation of patients for SS should appear in a separate subsection, and the Statistical analysis paragraph should appear as a subsection, not a separate section.

On page 6, the Authors should provide a brief description of the characteristic phenotypic abnormalities of T-LGL cells before describing immunophenotypes detected in the samples.

In Table 1, ‘no aberrant phenotype’ is indicated for patient #21. What does this mean?

On page 9, second paragraph, the word ‘justify’ seems inappropriate; ‘verify’ would be better.

In Figure 2, what does (Tube A) signify?

Reviewer 2 Report

The authors submitted short compressed data of an interesting observation that Sjogren's Syndrome (SS) is associated with T-LGL leukemia. The core of the manuscript is summarized in a table (Table 1) that includes read-out from multiple assays. 

The manuscript lacks a description of the methodology and experimental assays. 

Major comments:

1. Did the authors use whole blood or fresh PBMCs or frozen PBMCs for immunophenotyping? 

2. Why the Histology and immunologic assays were not separated in the method section? Instead, they were inserted in the FACS sort section.

3. In the result section, it is indicative that 2 out of the 21 samples had low flow signature/ND based on the table. But yet the rest of the interpretation included all 21. It is not clearly mentioned in the result section.

4. With regard to the flow data, can the authors kindly provide the gating strategies starting from total FSC/SSC plots? Cn the authors provide the freq or events for the 18 samples? Did the authors include FMOs control? For analyzing the CD4/CD8 low population, did the authors think of comparing the subsets using healthy donors' PBMCs as a control? 

5. Can the authors insert representative images of the molecular signature found with imaging? The readout from the table will be easier to interpret.

6. In the flow data, it would be better if the fonts of the axis labels were increased for visibility. 

7. To perform the T cell clonality, the authors use blood for DNA extraction. It would be better if the authors explain in brief the BIOMED-2 protocol.

8. Since CD4low is also CD4+, it would be better if the nomenclature is changed to CD4low and CD4high.

9. In figure 2, can the authors provide units for the y-axis? It is understood that the hotspot is in the 260bp range but it is not clear what the y-axis is.

10. For the antibody measurement, can the authors provide plots for anti-Ro/SSA?

Round 2

Reviewer 2 Report

Thank you to the authors for their patience and revision. The authors accepted the comments and made necessary corrections/insertions. 

I can understand the unavailability of samples for FMOs and healthy donors. 

Good luck! 
